# INTERACTIVE POST-TRAINING FOR VISION-LANGUAGE-ACTION MODELS

## ABSTRACT

We introduce `RIPT-VLA`, a simple and scalable reinforcement-learning-based inter-active post-training paradigm that fine-tunes pretrained Vision-Language-Action (VLA) models using only sparse binary success rewards. Existing VLA training pipelines rely heavily on offline expert demonstration data and supervised imita-tion, limiting their ability to adapt to new tasks and environments under low-data regimes. `RIPT-VLA` addresses this by enabling interactive post-training with a stable policy optimization algorithm based on dynamic rollout sampling and leave-on-out advantage estimation. `RIPT-VLA` has the following characteristics. First, `RIPT-VLA` applies to various VLA models, resulting in an improvement on the lightweight QueST model by 21.2%, and the 7B OpenVLA-OFT model to an unprecedented **97.5%** success rate. Second, `RIPT-VLA` is computationally efficient and data-efficient: With only one demonstration, `RIPT-VLA` enables an unworkable SFT model (**4%**) to succeed with a **97%** success rate within **15** iterations. Fur-thermore, we demonstrate that the policy learned by `RIPT-VLA` generalizes across different tasks and scenarios and is robust to the initial state context. These results highlight `RIPT-VLA` as a practical and effective paradigm for post-training VLA models through minimal supervision. Code and checkpoints will be released [1].

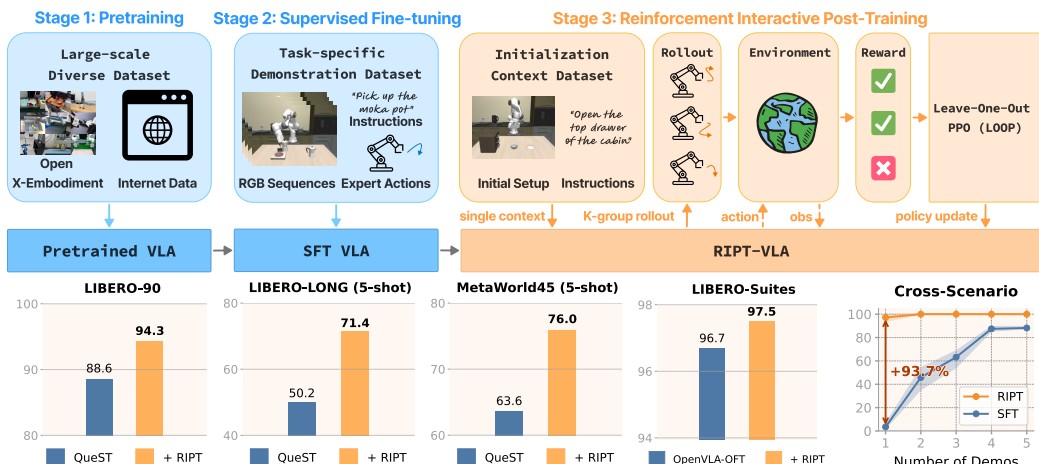

Figure 1: Overview of `RIPT-VLA`. While VLA models are typically trained with two supervised stages, we propose a third stage: Reinforcement Interactive Post-Training for VLA. `RIPT-VLA` sets state-of-the-art results across diverse benchmarks. It also presents remarkable improvement under low-data regime: transforms a 1-demo SFT model from near failure to 97% success.

## 1 INTRODUCTION

Vision-Language-Action (VLA) models (Zitkovich et al., 2023) aim to enable agents to perceive, reason, and act in the physical world with a unified interface. Current VLA models are trained

---

[1]We included anonymous code in the supplementary material for review.

with two supervised stages: large-scale pretraining on diverse human demonstrations, followed by supervised fine-tuning (SFT) on smaller-scale task-specific datasets. This paradigm has some distinct advantages: Pre-training enables the VLA model to build general visuomotor skills while SFT allows it to specialize in specific environments (Kim et al., 2024). Supervised training allows VLAs to learn from large-scale pre-recorded vision-language-action datasets. However, this supervised approach also has two core limitations: First, data is collected offline. The VLA learns to imitate interactions with the environment, but never sees the consequences of its own actions. The result is a policy often fails to handle the complexities of real-world scenarios, especially for long-horizon tasks. Second, task-specific SFT via imitation learning relies heavily on large-scale high-quality human demonstrations. This data is expensive and time-consuming to collect, and performance degrades significantly when only a small number of demonstrations are available.

In this paper, we propose `RIPT-VLA`: a third stage for VLA training paradigm with **R**einforcement **I**nteractive **P**ost-**T**raining. After pretraining and supervised fine-tuning, we allow the VLA model to interact with the multitask environment and receive binary success/failure rewards. We then optimize the VLA model to directly improve its success rate across multiple tasks through reinforcement learning. Inspired by prior RL frameworks for LLMs reasoning (Guo et al., 2025), we propose a stable and efficient RL framework for VLA finetuning in Section 4. Specifically, we extend the LOOP framework (Chen et al., 2025) which combines REINFORCE leave-one-out (RLOO) advantage estimation (Kool et al., 2019) and proximal policy optimization (PPO) (Schulman et al., 2017). Unlike LOOP, we construct uniform batches of non-zero advantage samples, filtering out any group of trajectories with zero-advantage, and sampling rollouts until sufficient samples exist. This uniform batch construction leads to improved training stability, especially as training progresses and the VLA becomes more successful. `RIPT-VLA` allows efficient and stable VLA policy update *without* relying on shaped or learned rewards, or critic models. Using Reinforcement Learning in a third training stage has a few distinct advantages: It is more data efficient, yielding close to state-of-the-art performance with only a single SFT demonstration. The resulting VLA has a much higher performance on the end-task, as it gets to see interactions with the environment during training. `RIPT-VLA` works with both tokenized (Mete et al., 2024) and continuous actions (Kim et al., 2025).

`RIPT-VLA` resonates with the recent trend of paradigm shift in LLM training (Guo et al., 2025). While pretraining on large-scale text corpora equips LLMs with broad knowledge and powerful skills, they often struggle with challenging tasks that require precise reasoning, multi-step planning, or tool use (Wang et al., 2024). To address these limitations, reinforcement learning has emerged as a critical third stage—used to reactivate and steer pretrained knowledge with only a small amount of interactive feedback (Ouyang et al., 2022). Similarly, we observe that pretrained VLA models also encode rich visuomotor skills, yet struggle to apply them effectively for new tasks and scenarios. `RIPT-VLA` bridges this gap by using only sparse binary rewards to unlock and specify these latent skills with a small number of optimization steps.

In Section 5, we demonstrate that `RIPT-VLA` achieves state-of-the-art results when combined with both large-scale and lightweight VLA models across a diverse set of tasks. On the LIBERO benchmark (Liu et al., 2023), `RIPT-VLA` improves QueST (Mete et al., 2024), the best lightweight VLA model, on all four task suites by **10.9%** absolute success rate (SR) on average (Table 1). When evaluated on OpenVLA-OFT (Kim et al., 2025), the best-performing large VLA model with an already high success rate (96.7%), `RIPT-VLA` still helps by further reducing the failure rate from 3.3% to 2.4%. We also achieve top performance on many-task benchmarks LIBERO-90 **(94.3%)** and MetaWorld45 (Yu et al., 2020) **(92.2%)**, showing the effectiveness of `RIPT-VLA` in improving multi-task (up to 90) performance with a single model (Table 2). Most notably, in the extreme low-data regime with only a single training demo, `RIPT-VLA` adapts pretrained knowledge to new tasks goals or scenarios with remarkable efficiency: boosting success rate from below 4% to over 97% within only 15 RL iterations.

## 2 RELATED WORKS

**Vision-Language-Action Models.** Vision-Language-Action (VLA) models empower embodied agents to interpret multimodal inputs—such as visual observations and natural-language instructions—and translate them into meaningful actions within the physical world (Zitkovich et al., 2023). Seminal works like RT-2 (Zitkovich et al., 2023), RT-1 (Brohan et al., 2022b), PaLM-E (Driess et al., 2023), Octo (Team et al., 2024), Dita (Hou et al., 2025), $\pi_0$ (Black et al., 2024), and $\pi_{0.5}$ (Intelligence et al., 2025), together with OpenVLA (Kim et al., 2024), showcase VLAs achieving emergent

semantic reasoning and generalization to novel tasks and environments. These models are typically developed through a two-stage supervised-learning paradigm that begins with an initial pre-training phase on extensive, web-scale datasets (Driess et al., 2023; Brohan et al., 2022a), which is crucial for acquiring generalizable visuomotor skills, grounding language in perception, and building robust internal representations. While this two-stage approach has advanced the field, its offline nature imposes key limitations. The supervised fine-tuning (SFT) stage typically requires vast expert demonstrations for new tasks or environments, thereby degrading few-shot performance. This highlights a critical gap: the need for methods that adapt pretrained VLAs beyond static imitation by leveraging interactive experience and reducing reliance on extensive expert data.

**Reinforcement Learning for LLMs.**   Large Language Models (LLMs) offer a precedent for enhancing pretrained models. While LLMs gain broad capabilities via pre-training and SFT, they often struggle with complex reasoning, planning, or constraint satisfaction (Wang et al., 2024). To address this, Reinforcement Learning (RL) has emerged as a transformative third stage in LLM training—enabling learning from interactive feedback rather than static datasets (Ouyang et al., 2022). Recent progress shows RL can unlock latent capabilities for math (Lightman et al.; Shao et al., 2024), self-verifiable proofs (Liu et al., 2024a), long-horizon planning through tree-of-thoughts (Yao et al., 2023), and preference-aligned generation with AI feedback (Lee et al., 2024a). This paradigm, in which pretrained knowledge is steered by targeted feedback, strongly motivates a similar approach for VLA models: RL has the potential to adapt pretrained VLAs more effectively to the interactive and consequential nature of embodied tasks.

**Reinforcement Learning for VLA.**   Recent works have explored applying reinforcement learning to pretrained VLA models to overcome limitations of supervised fine-tuning and adapt to novel tasks without collecting new demonstrations. iRe-VLA (Liu et al., 2024b) addresses optimization instability by alternating between PPO-based updates on a frozen VLM backbone and supervised distillation stages. However, it still relies on a learned value critic during PPO, and requires shaped reward functions or success weighting to guide policy learning. ConRFT (Ma et al., 2024) further combines offline Q-learning with online consistency-policy updates, but similarly depends on a parameterized value function. Both methods require careful coordination between offline and online stages to stabilize critic learning. In contrast, `RIPT-VLA` introduces a fully critic-free optimization framework with simpler training dynamics under sparse binary rewards.

## 3 PRELIMINARY

### 3.1 VISION-LANGUAGE-ACTION MODELS

**Autoregressive VLA rollout.**   A vision-language-action (VLA) model $\pi_\theta$ maps a sequence of observations and previous actions $(o_{1:t}, a_{1:t-1})$, along with a natural language goal $g$, to a probability distribution over the next action $a_t$. These models operate autoregressively: $a_t \sim \pi_\theta(\cdot \mid o_{1:t}, g, a_{1:t-1})$. Given an initial observation-goal pair context $\mathbf{c} = (o_1, g)$, the model generates a sequence of actions conditioned on past information in an autoregressive way: $\pi_\theta(a_{1:T} \mid o_{1:T}, g) = \prod_{t=1}^{T} \pi_\theta(a_t \mid o_{1:t}, g, a_{1:t-1})$. We denote this sampling process as $\mathbf{a} = a_{1:T} \sim \pi_\theta(\cdot \mid \mathbf{c})$, the observations alone the sequence as $\mathbf{o} = o_{1:T}$. Sequences terminate upon task success or reaching a time limit. For each rollout sequence and task goal $g$, the environment $\mathcal{E}$ returns a binary reward $R = 1$ when the task goal is successfully reached, and $R = 0$ otherwise. The environment $\mathcal{E}$ can be either a simulator (Yu et al., 2020; Liu et al., 2023) or the real world.

**Current VLA training paradigm.**   Current Vision-Language-Action (VLA) models are typically trained in two stages: **Stage 1: Pretraining** and **Stage 2: Supervised Fine-tuning**.

In Stage 1, a base policy $\pi_\theta$ is pretrained on a large-scale, diverse dataset of real-world demonstrations, denoted by $\mathcal{D}_{\text{pretrain}} = \{(\mathbf{o}, \mathbf{a}, g)\}_{i=1}^{N}$. The policy is trained to imitate the ground-truth actions given offline data in $\mathcal{D}_{\text{pretrain}}$. For VLA with tokenized action head, the loss is:

$$\mathcal{L}_{\text{pre}}(\theta) = -\mathbb{E}_{(\mathbf{o}, \mathbf{a}, g) \sim \mathcal{D}_{\text{pretrain}}} \left[ \sum_{t=1}^{T} \log \pi_\theta(a_t \mid o_{1:t}, g, a_{1:t-1}) \right], \qquad (1)$$

while for regression action head $\mathcal{L}_{\text{pre}}(\theta)$ is implemented as an MSE or L1 loss. This stage enables VLA capture strong representations and learn general visuomotor and instruction-following capabilities.

In Stage 2, the pretrained policy is supervised fine-tuned on a smaller, multitask dataset to improve performance on a small set of target tasks, denoted by $\mathcal{D}_{\text{sft}} = \{(\mathbf{o}, \mathbf{a}, g)\}_{i=1}^{N'}$. Typically, $\mathcal{D}_{\text{sft}}$ contains around 50 high-quality human demonstrations per task (Mete et al., 2024). The VLA is trained with the same objective function as in Stage 1. This stage enables the model to adapt its learned skills from Stage 1 to a specialized set of skills for the target tasks.

Although being the standard process of VLA training, this two-stage process has two significant issues. Firstly, it relies only on offline supervision and lack interactive feedback from the environment. Therefore, the learned policy may often fail in real rollouts due to distribution shift and cascading errors, especially for long-term rollout. Furthermore, the performance of VLA heavily relies on the high quality and quantity of the task-specific data in $\mathcal{D}_{\text{sft}}$, which is often hard and costly to obtain.

**VLA as Markov decision processes.** To better optimize VLA models, we define its task as a Markov decision process (MDP). Each episode is initialized with a context $\mathbf{c} = (o_1, g)$. The *state* is represented as $[o_{1:t}, g, a_{1:t-1}]$, which includes the language goal $g$, the sequence of past observations $o_{1:t}$, and past actions $a_{1:t-1}$. At each timestep $t$, the VLA policy produces an *action* sampled from the policy distribution: $a_t \sim \pi_\theta(\cdot \mid o_{1:t}, g, a_{1:t-1})$. The environment transitions to the next observation $o_{t+1}$ based on hidden environment dynamics, producing a new state $[o_{1:t+1}, g, a_{1:t}]$. After a sequence of actions $a_{1:T}$, the agent receives a binary *reward* $R(\mathbf{c}, \mathbf{a}) \in \{0, 1\}$ from the environment $\mathcal{E}$, indicating task success or failure. The objective of VLA optimization is essentially learning a policy $\pi_\theta$ that maximizes expected task success reward: $L_\theta(\mathbf{c}) = \mathbb{E}_{\mathbf{a} \sim \pi_\theta(\cdot|\mathbf{c})}[R(\mathbf{c}, \mathbf{a})]$.

### 3.2 REINFORCEMENT POLICY OPTIMIZATION

We consider the reinforcement learning setting where an agent interacts with an environment $\mathcal{E}$ to learn a policy $\pi_\theta(\mathbf{a} \mid \mathbf{c})$ that maximizes the expected return: $\mathbb{E}_{\mathbf{c} \sim \mathcal{D}_{\text{context}}, \mathbf{a} \sim \pi_\theta}[R(\mathbf{c}, \mathbf{a})]$, where $\mathbf{c}$ is the context (e.g., goal and initial observation), $\mathbf{a}$ is a trajectory (e.g., sequence of actions), and $R(\mathbf{c}, \mathbf{a}) \in \{0, 1\}$ is a sparse binary reward returned by the environment. To optimize this objective, a standard approach is policy gradient, which updates $\pi_\theta$ with: $\nabla_\theta L_\theta(\mathbf{c}) = \mathbb{E}_{\mathbf{a} \sim \pi_\theta}[\nabla_\theta \log \pi_\theta(\mathbf{a} \mid \mathbf{c}) \cdot A(\mathbf{c}, \mathbf{a})]$. Here, $A(\mathbf{c}, \mathbf{a})$ is the advantage function indicating how much better the action $\mathbf{a}$ is compared to a baseline. In practice, computing $A(\mathbf{c}, \mathbf{a})$ is challenging, especially under sparse rewards. To address this issue, a recent work proposed a critic-free optimization framework called Leave-One-Out Proximal Policy Optimization (**LOOP**) (Chen et al., 2025). It combines the two methods below:

**Leave-One-Out Advantage Estimation (RLOO) (Kool et al., 2019).** For each sampled context $\mathbf{c}$, we draw $K$ rollouts $\{\mathbf{a}_k \sim \pi_\psi(\cdot \mid \mathbf{c})\}_{k=1}^K$ under a fixed sampling policy $\pi_\psi$. Each rollout receives a binary reward $R_k = R(\mathbf{c}, \mathbf{a}_k)$. The leave-one-out baseline for rollout $k$ is computed by averaging the rewards from all the other rollouts:

$$b_k = \frac{1}{K-1} \sum_{j \neq k} R_j, \quad A_k = R_k - b_k. \tag{2}$$

This group-normalized advantage indicates how much better or worse a rollout performance relative to others from the same context. This allows use to efficiently compute a stable advantage signal from sparse binary rewards, without requiring learning value functions.

**Proximal Policy Optimization (PPO) (Schulman et al., 2017).** To update $\pi_\theta$ using collected rollouts $\{(\mathbf{c}_k, \mathbf{a}_k, A_k)\}$, we compute the importance ratio $r_k = \pi_\theta(\mathbf{a}_k \mid \mathbf{c}_k)/\pi_\psi(\mathbf{a}_k \mid \mathbf{c}_k)$, where $\pi_\theta$ is the current updating policy and $\pi_\psi$ is the fixed sampling policy (normally set to the last checkpoint of $\pi_\theta$). We then optimize $\pi_\theta$ with the following clipped objective (PPO loss): $\mathcal{L}_{\text{PPO}} = -\min(r_i A_i, \text{clip}(r_i, 1 - \epsilon, 1 + \epsilon) A_i)$. Here, $\epsilon$ is a small updating threshold (we set to 0.2). This objective encourages rollouts with positive advantages while preventing unstable updates when $\pi_\theta$ deviates too far from its previous version $\pi_\psi$.

LOOP adopts PPO to optimize the advantage estimated by RLOO, which enables sample-efficient policy optimization in sparse reward settings without critics. It serves as an out-of-box working implementation for our interactive post-training framework in Section 4.

## 4 RIPT-VLA

As mentioned above, there is a gap between the current VLA training paradigm and our essential goal of making it work in our downstream tasks. On one hand, pure supervised training on offline data makes the policy fragile in real rollout due to compounding errors and the distribution gap between

---

**Algorithm 1 RIPT-VLA**: **R**einforcement **I**nteractive **P**ost-**T**raining for **VLA** Model

---

**Input:** Pretrained VLA $\pi_\theta$, reward function $R(\mathbf{c}, \mathbf{a})$,

1: context dataset $\mathcal{D}_{\text{context}}$
2: **for** step $= 1$ to $M$ **do**
3:     Update sampling VLA $\pi_\psi \leftarrow \pi_\theta$
4:     Initialize empty dataset $\mathcal{D}_{\text{rollout}} \leftarrow \emptyset$
5:     **while** $|\mathcal{D}_{\text{rollout}}| < B$ **do**
6:         Sample a context $\mathbf{c} \leftarrow (g, o_1) \sim \mathcal{D}_{\text{context}}$
7:         Generate $K$ rollouts $\{\mathbf{a}_k \sim \pi_\psi(\cdot \mid \mathbf{c})\}_{k=1}^K$
8:         Compute rewards $\{R_k \leftarrow R(\mathbf{c}, \mathbf{a}_k)\}_{k=1}^K$
9:         Compute baselines: $b_k \leftarrow \frac{1}{K-1} \sum_{j \neq k} R_j$
10:       Compute advantages: $A_k \leftarrow R_k - b_k$ for each $k$
11:       **if** all $A = 0$ **then**
12:           **continue**
13:       **end if**
14:       Add $(\mathbf{c}, \mathbf{a}_k, A_k)$ for all $k$ to $\mathcal{D}_{\text{rollout}}$
15:     **end while**
16:     **for** iteration $= 1$ to $N$ **do**
17:         Update $\pi_\theta$ with PPO loss over $\mathcal{D}_{\text{rollout}}$
18:     **end for**
19: **end for**

---

offline dataset and online rollout. Furthermore, one has to collect a sufficient number of high-quality demonstrations for the offline datasets, especially $\mathcal{D}_{\text{sft}}$, the model can easily overfit to the training distribution. In other words, optimizing VLA through Equation 1 does not necessarily improve the VLA's task execution success rate. To bridge this gap, we propose a new VLA training paradigm that directly optimize pretrained VLA through interaction with the environment $\mathcal{E}$ through **R**einforcement **I**nteractive **F**ine-**T**uning. We call this paradigm RIPT-VLA.

### 4.1 INTERACTIVE POST-TRAINING FOR VLA

The first two stages of our VLA training paradigm are the same as standard setting. In Stage 1, We pretrain the VLA model on a large diverse dataset $\mathcal{D}_{\text{pretrain}}$ to learn visual-language representation and general visuomotor skills. Then, in Stage 2 we finetune VLA on a small dataset $\mathcal{D}_{\text{sft}}$ to adapted it to follow instructions to solve a small set of target tasks. These stages produce a pretrained VLA policy $\pi_\theta$ that can achieve non-zero success rate (can be very low) on the target tasks.

In RIPT-VLA, we then conduct **Stage 3: Reinforcement Interactive Post-Training**. In this stage we assume we can rollout $\pi_\theta$ in an environment $\mathcal{E}$ and receive a binary reward $R(\mathbf{c}, \mathbf{a}) \in \{0, 1\}$ given $\mathbf{a} \sim \pi_\theta(\cdot \mid \mathbf{c})$, where $\mathbf{c}$ is the initial context. In addition, we use an initial context dataset $\mathcal{D}_{\mathbf{c}} = \{(o_1, g)\}$ to set up task initializations for model rollouts. Typically, we obtain $\mathcal{D}_{\mathbf{c}}$ by directly extracting the initial states from sequences in $\mathcal{D}_{\text{sft}}$. For each optimization step, we iterate between two steps: **rollout collection** and **policy optimization**.

During *rollout collection*, we randomly sample contexts $\mathbf{c}_i \sim \mathcal{D}_{\mathbf{c}}$ and let $\pi_\theta$ interact with the environment $\mathcal{E}$ to output a sequence $\mathbf{a}_i$. For each rollout we collect its reward $R(\mathbf{c}_i, \mathbf{a}_i)$ and compute its advantage $A_i = A(\mathbf{c}_i, \mathbf{a}_i)$, which indicate how strong the model should be encouraged ($A > 0$) or penalized ($A < 0$) for generating rollout $\mathbf{a}$. We add all rollouts and rewards $(\mathbf{c}_i, \mathbf{a}_i, A_i)$ to a rollout dataset $\mathcal{D}_{\text{rollout}}$ until we obtain $B$ rollouts: $\mathcal{D}_{\text{rollout}} = \{(\mathbf{c}_i, \mathbf{a}_i, A_i)\}_{i=1}^B$

During *policy optimization*, we optimize $\pi_\theta$ with reinforcement learning algorithms on $\mathcal{D}_{\text{rollout}}$ to maximize its expected task success rate for $N$ iterations. After optimization, we use the updated VLA policy $\pi_\theta'$ to collect new rollouts and a new step begins. This process repeats until we reach $M$ steps and outputs the final policy $\pi_\theta^*$, concluding the full VLA training paradigm. We then deploy $\pi_\theta^*$ in the environment for testing.

Although RIPT-VLA is simple in concept, it presents several challenges. First, we only have sparse binary rewards from each rollout sequence, no shaped reward is available. Training a learned reward model to predict shaped reward values can easily lead to reward hacking (Skalse et al., 2022),

especially with limited rollout data. Second, as VLA models operate over long-horizon, multi-task environments, credit assignment becomes highly ambiguous. This causes the value target (e.g., from TD error) to be extremely noisy and uninformative. Third, training a stable value function for VLA requires a model of comparable capacity to the VLA itself, which significantly increases GPU memory usage and training cost for large VLA models (Zhai et al., 2024). Finally, in multitask environments, different task contexts can vary significantly in difficulty: some lead to trivial success while others consistently fail across all rollouts. This results in highly imbalanced success rates and unstable policy gradient updates.

## 4.2 DYNAMIC-SAMPLING LEAVE-ONE-OUT PROXIMAL POLICY OPTIMIZATION.

To implement `RIPT-VLA` in a stable and sample-efficient way, we propose a simple yet effective policy optimization framework in Algorithm 1. First, we adopt LOOP (Section 3.2) as the foundation of our implementation. LOOP is particularly well-suited for our VLA setting, where rollouts are long-horizon and efficient advantage estimation is required for its sparse reward signal. Furthermore, for VLA in multitask environments, we design a dynamic rollout sampling mechanism to filter out uninformative contexts for more stable and efficient policy optimization.

**LOOP for `RIPT-VLA`.** We apply LOOP (Chen et al., 2025) for both the rollout collection and policy optimization stage. During rollout collection, we conduct RLOO (Kool et al., 2019) advantage estimation. In this step, we use the most recent policy $\pi_\theta$ as the sampling policy $\pi_\psi$. Given a single context $\mathbf{c} \sim \mathcal{D}_\mathbf{c}$, we collect $K$ trajectories by repeatedly sampling $K$ times from the policy given the same context: $\{\mathbf{a}_k \sim \pi_\psi(\cdot \mid \mathbf{c})\}_{k=1}^K$. We obtain their corresponding rewards $\{R_k\}_{k=1}^K$ from the environment $\mathcal{E}$. For each rollout $k$, we compute the advantage $A_k$ with Equation 2. For each epoch, we conduct group sampling on $B/K$ contexts sampled from $\mathcal{D}_\mathbf{c}$, obtaining $\mathcal{D}_\text{rollout}$ with $B$ rollouts.

During policy optimization, we use PPO (Schulman et al., 2017) to stabilize policy gradient updates. For each rollout sample $(\mathbf{c}_i, \mathbf{a}_i, A_i) \in \mathcal{D}_\text{rollout}$, we can compute its training objective $\mathcal{L}_\text{PPO}$. We perform this update over the collected rollout dataset $\mathcal{D}_\text{rollout}$ using mini-batches for $N$ optimization steps each epoch. When $N = 1$, the method corresponds to on-policy RLOO; when $N > 1$, the same samples are reused for additional updates, resulting in a partially off-policy optimization.

**Dynamic rollout sampling.** VLA models often operate in multitask environments (Kim et al., 2024; Mete et al., 2024; Sun et al., 2022), where task difficulty varies widely across different contexts. Some contexts have been already well solved by VLA, leading to trivial success across $K$-group sampling, while others consistently fail due to inherent task complexity or distribution gap. Both cases result in rollout groups where all rollout samples receive identical rewards (all 1s or all 0s), producing all 0 advantage in Equation 2. Therefore there is no gradient signal from the PPO loss $\mathcal{L}_\text{PPO}$. Adding these samples to $\mathcal{D}_\text{rollout}$ makes unstable gradient updates during batch optimization, as they contribute zero gradients that can dominate or dilute meaningful learning signals.

To address this, we apply a simple yet effective dynamic rejection strategy: we discard any sampled context for which all $K$ rollouts receive the same reward and resample a new context from $\mathcal{D}_\text{context}$ for group sampling. As training progresses and the policy improves, an increasing number of task contexts yield uniformly successful rollouts. Dynamic rejection naturally filters out these solved contexts, allowing optimization to concentrate on the remaining harder contexts. Importantly, this method make the batch optimization of $\mathcal{L}_\text{PPO}$ to have the same effective batch size over all the minibatches across $\mathcal{D}_\text{rollout}$, which we empirically found to be important for stable policy optimization in `RIPT-VLA`.

## 4.3 GENERALIZE TO DIFFERENT VLA MODELS.

`RIPT-VLA` is compatible with both discrete and continuous action representations commonly used in VLA models. To perform stable policy optimization, we compute the trust region $r_i = \frac{\pi_\theta(\mathbf{a}_i|\mathbf{c}_i)}{\pi_\psi(\mathbf{a}_i|\mathbf{c}_i)}$ in the PPO loss to constrain policy updates within a small region of the original policy. A key component in this formulation is computing the log-probability of the sampled action sequences under both policies. At each step, we assume the policy outputs a probability distribution over actions. We compute the log-probability of a sampled action sequence $\mathbf{a} = (a_1, \dots, a_T)$ as the sum of the per-step log-probabilities: $\log \pi_\theta(\mathbf{a} \mid \mathbf{c}) = \sum_{t=1}^T \log \pi_\theta(a_t \mid a_{<t}, \mathbf{c})$. Therefore, we can apply `RIPT-VLA` to any VLA model $\pi_\theta$ that we can compute $\log \pi_\theta(a_t \mid a_{<t}, \mathbf{c})$.

**Tokenized action head.**   For VLA models with discrete action outputs, *e.g.* QueST (Mete et al., 2024), actions are predicted as sequences of discrete tokens from a fixed vocabulary, where the action header is a classification head trained with NLL loss. Therefore, $\log \pi_\theta(a_t \mid a_{<t}, \mathbf{c})$ is directly obtained from applying softmax function to the model's classification head output logits.

**Regression action head.**   For continuous-action VLA models (Kim et al., 2025), actions are regressed using MSE or L1 loss, which do not produce a log-probability. To enable policy gradient optimization, we extend the model with a light-scale prediction head that estimates the scale $\sigma_\theta$ of the action value. Assuming the original output head provides the mean $\mu_\theta$, we treat the policy as a factorized Gaussian (MSE) or Laplace (L1) distribution and train the scale head using the NLL loss in Equation 1 for a few iterations on $\mathcal{D}_{\text{sft}}$. After that, we can compute $\log \pi_\theta(a_t \mid a_{<t}, \mathbf{c})$ with predicted $\mu_\theta$ and $\sigma_\theta$ in a closed form.

## 5 EXPERIMENTS

We evaluate `RIPT-VLA` on two widely used benchmarks for VLA learning: LIBERO (Liu et al., 2023) and MetaWorld (Yu et al., 2020). We study several settings: (1) standard multitask (up to 90 tasks) setting in Sec. 5.2, (2) few-shot ($1 \sim 5$ demonstration) setting in Sec. 5.3, and (3) cross-task and cross-scenario setting in Secs. A.1 and A.2 to showcase the ability of fast generalization leveraging prior knowledge during pretraining. Additionally, we additional studies to analyze the practical behavior of `RIPT-VLA`, including training curves, ablation studies as well as its sensitivity to the variance and diversity of the context dataset.

### 5.1 SETUP

**Benchmark.**   LIBERO (Liu et al., 2023) is a lifelong learning benchmark with 5 task suites. Each suite consists of a set of language-guided manipulation tasks across multiple object types, task definitions and environment scenarios. Specifically, it includes 4 suites: **Goal**, **Spatial**, **Object**, and **Long**. Each suite is designed to evaluate a specific aspect of object manipulation and containing 10 distinct tasks. In addition, it also includes a **LIBERO-90** suite that contains 90 different tasks to access multitask performance at scale. MetaWorld (Yu et al., 2020) is a manipulation task benchmark for few-shot learning models. We use Meta-Learning 45 (ML45) suite that contains 45 training tasks and 5 held-out tasks.

For both benchmarks, each task comes with 50 expert demonstrations for training. At evaluation time, a single VLA model is deployed across all tasks in a suite and performs rollouts on 50 held-out test contexts per task. We measure performance with the average task success rate (SR).

**Base models.**   We conduct `RIPT-VLA` on two pretrained VLA with different design choices.

OpenVLA-OFT (Kim et al., 2025) is an *Optimized Fine-Tuned* variant of the 7B OpenVLA model (Kim et al., 2024). OpenVLA is initialized from a multimodal backbone that combines a *Llama-2 7B* language model with dual vision encoders (Oquab et al., 2023; Zhai et al., 2023) and is pretrained on 970k robot-manipulation demonstrations. OFT replaces the original tokenized action decoder with a continuous decoding head and trains with an L1 regression loss. This architecture represents the *large-scale regression action* VLA. QueST (Mete et al., 2024) on the other hand, is a *small-scale tokenized action* VLA model with 20 million parameters. QueST first learns a VQ-VAE that compresses short motion segments into a discrete *skill codebook*; a GPT-style transformer then autoregressively predicts these skill tokens conditioned on images and language, and a small decoder turns tokens back into continuous joint commands.

**Implementation details.**   We implement `RIPT-VLA` with method described in Section 4.2. Unless otherwise specified, we construct $\mathcal{D}_{\mathbf{c}}$ from all initial states in the supervised fine-tuning dataset $\mathcal{D}_{\text{sft}}$. For OpenVLA-OFT, we finetune the model from official checkpoints for each task suite. We train on 4 RTX A5000 GPUs using LoRA (Hu et al., 2022) with rank 32 on 4 GPUs, and set $K = 8$, $B = 192$, $N = 1$ and $\epsilon = 0.1$. We set a learning rate of $1\mathrm{e}{-4}$ for the LoRA modules and $1\mathrm{e}{-5}$ for the action head. Following Section 4.3, before applying `RIPT-VLA`, we first train a small Laplace scale header from scratch (with the same architecture as the action header) with NLL loss on $\mathcal{D}_{\text{sft}}$ for 500 steps.

For QueST, as official checkpoints are not provided, we first train the base model from scratch for each task suite following the official code and hyper-parameters. In the multitask setting, we conduct

| Stage 1 + Stage 2 Models | | | | | |
|---|---|---|---|---|---|
| **Method** | **Goal** | **Spatial** | **Object** | **Long** | **Average** |
| Octo (Team et al., 2024) | 84.6 | 78.9 | 85.7 | 51.1 | 75.1 |
| OpenVLA (Kim et al., 2024) | 79.2 | 84.7 | 88.4 | 53.7 | 76.5 |
| Dita (Hou et al., 2025) | 85.4 | 84.2 | 96.3 | 63.8 | 82.4 |
| $\pi_0$ + FAST (Pertsch et al., 2025) | 88.6 | 96.4 | 96.8 | 60.2 | 85.5 |
| $\pi_0$ (Black et al., 2024) | 95.8 | 96.8 | **98.8** | 85.2 | 94.2 |
| OpenVLA-OFT* (Kim et al., 2025) | 97.9 | 97.6 | 98.4 | 92.9 | 96.7 |
| **OpenVLA-OFT + RIPT** | **99.0** | **98.6** | 98.6 | **93.8** | **97.5** |
| (improvement) | **(+1.1)** | **(+1.0)** | **(+0.2)** | **(+0.9)** | **(+0.8)** |
| Stage-2 Models | | | | | |
| **Method** | **Goal** | **Spatial** | **Object** | **Long** | **Average** |
| Diffusion Policy (Chi et al., 2023) | 68.3 | 78.3 | 92.5 | 50.5 | 72.4 |
| Seer (Tian et al., 2024) | – | – | – | 78.7 | – |
| MDT (Reuss et al., 2024) | 73.5 | 78.5 | 87.5 | 64.8 | 76.1 |
| MDT+ (Reuss et al., 2024) | – | 95.2 | 97.8 | 83.0 | – |
| QueST (Mete et al., 2024) | 80.8 | 87.4 | 93.6 | 68.8 | 82.7 |
| **QueST + RIPT** | **92.7** | **95.6** | **98.4** | **87.5** | **93.6** |
| (improvement) | **(+11.9)** | **(+8.2)** | **(+4.8)** | **(+18.7)** | **(+10.9)** |

Table 1: Multitask SR(%) on the four LIBERO suites. **Bold** indicates best result and underline marks the second-best. Improvements from `RIPT-VLA` are **marked in red**. *: OpenVLA-OFT results are obtained from running official checkpoints for each suite.

`RIPT-VLA` on 3 GPUs with $K = 16$, $B = 2880$ ($16 \times 180$). For single-task setting, we use 1 GPU with $K = 16$, $B = 160$. For both settings, we set $N = 20$, PPO mini-batch size as 24, a learning rate of 1e−6, and the clipping parameter $\epsilon = 0.2$. Please refer to the Appendix for more details.

## 5.2 STANDARD MULTITASK TRAINING

In this section we evaluate `RIPT-VLA` under standard multitask benchmarks. For each suite we use all the 50 expert demonstrations per task as its SFT dataset $\mathcal{D}_{\text{sft}}$. We conduct `RIPT-VLA` to finetune a base model on the corresponding dataset for each task suite.

Table 1 compares multitask performance on four LIBERO suites for different VLA models. We organize the results into two sets based on VLA training paradigm. In the **Stage 1+ Stage 2** set, we include 5 state-of-the-art large VLA models, which are typically larger than 500M parameters, pretrained (Stage-1) on large-scale general-purpose datasets, *e.g.,* Open-X Embodiment (O'Neill et al., 2024), and then finetuned using 50 demonstrations per task for each LIBERO suite (Stage-2). In contrast, the **Stage 2** set includes 4 representative small models, which are within 50M parameters and are directly trained on each LIBERO suite from scratch.

We show that `RIPT-VLA` significantly improves the best-performing VLA model in both types, setting new state-of-the-art performance on the 4 LIBERO suites. Specifically, `RIPT-VLA` improves QueST on all four task suites by **10.9** absolute SR on average, and yields even larger gains of **18.7** for the challenging LONG suite. Notably, with `RIPT-VLA`, the small 20M QueST model achieves much better performance with large models like Dita (334M) and comparable with $\pi_0$ (2B). When applying to OpenVLA-OFT, the best-performing large VLA model with already high SR, `RIPT-VLA` still further reduces the average failure rate from 3.3% to 2.5%. By applying `RIPT-VLA`, we set new state-of-the-art performance on 3 out of the 4 LIBERO suites (with only a 0.2 gap on the Object suite), and achieve the highest average success rate across all tasks. These results show the `RIPT-VLA` is broadly effective: it can both unlock latent capabilities in small-scale models and further push the limits of the high-performing ones.

In addition, in the left two columns of Table 2, we show the results on LIBERO-90 and ML45, which contain 90 and 45 diverse tasks respectively. These benchmarks assess the scalability and generalization of a single VLA model across many skills. We apply `RIPT-VLA` to QueST and compare with representative imitation learning methods: ACT (Gao et al., 2024), PRISE (Zheng et al., 2024), Diffusion Policy (Chi et al., 2023), VQ-BeT (Lee et al., 2024b) and ResNet-T (Mete et al., 2024). We

| Method | Full Data | | 5-shot Data | |
|---|---|---|---|---|
| | LIBERO-90 | ML45 | LONG | ML45 |
| ACT (Gao et al., 2024) | 50.8 | 90.8 | 42.0 | 70.8 |
| PRISE (Zheng et al., 2024) | 54.4 | 80.4 | 52.7 | 66.8 |
| DP (Chi et al., 2023) | 75.4 | 90.3 | 45.9 | 65.0 |
| VQ-BeT (Lee et al., 2024b) | 81.3 | 87.6 | 41.8 | 65.6 |
| ResNet-T (Mete et al., 2024) | 84.4 | 88.4 | 51.9 | 54.0 |
| QueST (Mete et al., 2024) | 88.6 | 91.0 | 50.2 | 63.6 |
| **QueST + RIPT** | **94.3** | **92.2** | **71.4** | **76.0** |
| (improvement) | (+5.7) | (+1.2) | (+21.2) | (+12.4) |

Table 2: Mean Success Rate (SR%) across four evaluation settings.    Figure 2: Few-shot curve.

show that `RIPT-VLA` improves performance of QueST by **5.7** and **1.2** absolute SR for LIBERO-90 and ML45, again setting new SOTA performance for both benchmarks. This confirms the utility of `RIPT-VLA` not only for improving performance on a few related task, but also scale up to broader, more realistic scenarios where a single model solving many different tasks.

### 5.3 FEW-SHOT MULTITASK TRAINING

In this section we evaluate `RIPT-VLA` under few-shot multitask setting. For each suite, we uniformly sample 1 to 10 expert demonstrations from each task to constitute the few-shot SFT dataset $\mathcal{D}_{\text{sft}}$. This setting reflects practical situation where large-scale data collection is not available.

The right two columns of Table 2 show results under the 5-shot setting, where each task in the LIBERO-LONG and ML45 suites is trained with only 5 demonstrations. While baseline models struggle in this low-data regime, `RIPT-VLA` significantly improves QueST by **21.2** on LIBERO-LONG and **12.4** on ML45. These results demonstrate that `RIPT-VLA` effectively addresses a key limitation of standard VLA training with SFT: it enables strong performance even with minimal demonstrations, alleviating concerns about data scarcity in real-world multitask deployment.

To further investigate the effect of the number of few-shot demonstrations, we conduct experiments under varying few-shot settings with QueST, ranging from 1 to 10 demonstrations per task on LIBERO-LONG. As shown in Figure 2, `RIPT-VLA` consistently largely improve the performance of standard SFT model across all data scales. Note that even for the extremely low-data regime, where we only have 1 demonstration per task, `RIPT-VLA` can still acehive a **20.8** absolute gain. As the number of demonstrations increases, `RIPT-VLA` continues to yield performance improvements, indicating its strong sample efficiency and scalability. These results confirm that `RIPT-VLA` is robust across different levels of data scarcity and is applicable in both low- and high-data settings.

### 5.4 ADDITIONAL RESULTS

We further include a series of extensive studies of `RIPT-VLA` in the Appendix. First, we investigate **cross-scenario** (Appendix A.1) and **cross-goal** (Appendix A.2) generalization, showing that `RIPT-VLA` enables pretrained visuomotor skills to transfer effectively across environments and task semantics under extremely low-shot supervision. In Appendix A.3, we provide ablation studies on our dynamic rollout sampling strategy, the size of the context dataset, and the effect of rollout variance. We encourage readers to refer to the Appendix for these interesting experiments and deeper analysis.

## 6 CONCLUSION

We presented **RIPT-VLA**, a simple yet powerful reinforcement learning paradigm for post-training pretrained Vision-Language-Action (VLA) models using sparse binary task rewards. `RIPT-VLA` enables stable and data-efficient optimization without the need for shaped rewards, value functions, or reward modeling. Our method significantly improves performance across multiple VLA benchmarks, and demonstrates remarkable adaptability even in extremely low-data settings. `RIPT-VLA` serves as a scalable third-stage training paradigm that complements existing pretraining and supervised fine-tuning pipelines, unlocking the latent potential of large VLA models through direct environment interaction. An exciting future direction is to combine `RIPT-VLA` with reasoning and planning in VLA models to enable more sophisticated and generalizable behaviors in complex environments.

## 7 ETHICS STATEMENT

Our experiments are conducted entirely in simulation using publicly available benchmarks (e.g., LIBERO suites, ML45), and do not involve human subject data, personal information, or sensitive attributes. The proposed method aims to advance sample-efficient reinforcement learning for Vision-Language-Action (VLA) models. While embodied AI carries potential downstream risks if misused, our study is limited to controlled settings. We encourage researchers extending this work to follow established safety protocols when transferring models to physical systems. We acknowledge the environmental impact of training large models. Our approach is computationally lightweight compared to training VLAs from scratch, as it refines existing pretrained models with modest reinforcement learning iterations. To ensure integrity and reproducibility, we disclose our use of large language models (LLMs) for writing assistance in Appendix, and provide full experimental details, code, and scripts in the supplementary materials.

## 8 REPRODUCIBILITY STATEMENT

We provide an anonymous code release in the supplementary material, including detailed installation instructions and ready-to-run scripts for reproducing all experiments. All hyperparameters, model architectures, and training protocols are described and specified in the corresponding configure files. The datasets and benchmarks used (LIBERO suites, and ML45) are publicly available. Together, these resources ensure that our results can be reliably reproduced and extended.

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

# A APPENDIX

## A.1 CROSS-SCENARIO GENERALIZATION

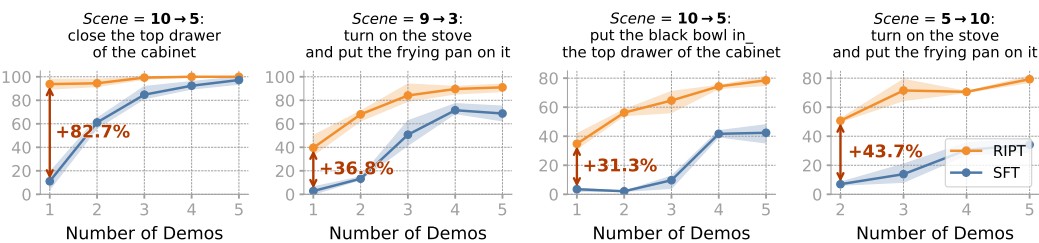

Figure 3: Cross-scenario task generalization from Scenario A to Scenario B with the same goal.

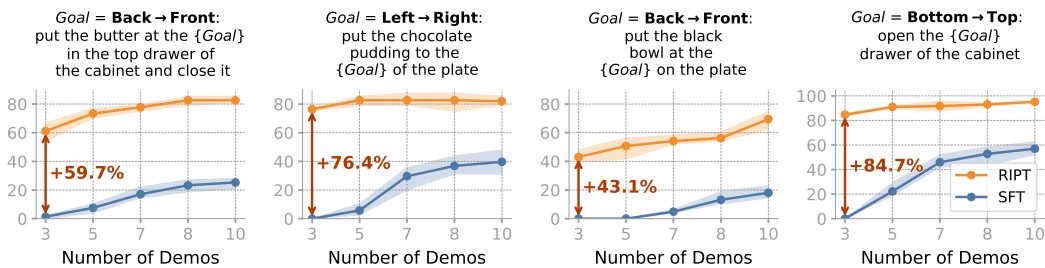

Figure 4: Cross-goal task generalization from Goal A to Goal B in the same scenario.

Recent paradigm shift in LLM training demonstrate that reinforcement learning can reactivate and steer pretrained knowledge with only a small amount of interactive feedback (Ouyang et al., 2022). We adopt a similar approach for VLA and ask: can RIPT-VLA enable sample-efficient pretrained visuomotor skill transfer across scenarios and goals?

In this section, we conduct experiment on the few-shot cross-scenario generalization setup. For each experiment, we consider a pair of tasks that have the same taks goal (e.g., *'turn on the stove and put the frying pan on it'*), but operate in different scenarios: Scenario A and Scenario B - with distant background layouts and object configurations. In Stage 1, we pretrain QueST on $|\mathcal{D}_{\text{pretrain}}| = 50$ demonstrations from Scenario A to acquire general visuomotor skill for this task goal. In Stage 2, we conduct SFT on $|\mathcal{D}_{\text{sft}}| = \{1, 2, 3, 4, 5\}$ demonstrations from Scenario B. Then, in Stage 3 we apply RIPT-VLA to optimize the policy through interactive rollouts on contexts $\mathcal{D}_{\text{context}}$ extracted from $\mathcal{D}_{\text{sft}}$. We then evaluate the model performance on the 50 testing contexts of Scenario B. We conduct experiments with 3 random seeds and plot the mean and variance across different $\mathcal{D}_{\text{sft}}$ sizes.

Figure 3 and Figure 1 right [2] show results on 5 scenario pairs. We observe that standard SFT on VLA models clearly struggles in the **1-shot** regime, achieving an average success rate of only around 5%, and in some cases dropping as low as 2%. Clearly, SFT fails to generalize the task knowledge from the pretraining stage to the new scenario. In contrast, RIPT-VLA dramatically improves performance, with absolute SR gain as high as **93.7%** (from 3.5% SFT to 97.2%). As the size of $\mathcal{D}_{\text{sft}}$ increases, both SFT and RIPT-VLA performance improve, but RIPT-VLA consistently maintains a strong improvement, often reaching near-100% performance with just 3-5 demonstrations. These results supports our core assumption: RIPT-VLA enables pretrained VLA models to rapidly activate and adapt learned skills with sparse binary rewards feedbacks.

## A.2 CROSS-GOAL GENERALIZATION

In this section, we investigate RIPT-VLA in a cross-goal generalization setting. Here we focus on task pairs that operate in the same scenario but different goals. Specifically, we select Task A and

---

[2]Curve setup: A= Scenario 5, B = Scenario 10, Task="close the top drawer of the cabinet".

Task B such that they require the same visuomotor skills but have different goals. For example, Task A is *"put the red mug on the **right** plate"* while Task B is *"put the red mug on the **left** plate"*. This setting tests whether pretrained visuomotor primitive skills (e.g., pick up and move) can be reused and recomposed to solve novel task goals (e.g., left vs. right). We again follow the 3 Stage paradigm: pretrain QueST on 50 demonstrations of Task A, SFT on a 3-10 demonstrations on Task B, and then apply RIPT-VLA for Task B.

Figure 4 presents result over 5 set of tasks. We observe that cross-goal generalization is significantly more challenging. With 3 demonstrations, SFT models still struggles and reach only **0.7%** success rate on average, almost not workable at all. With RIPT-VLA, we can improve model performance to **59.7%** on average. Remarkably, for one task pair, RIPT-VLA improves the performance from near **0%** success rate to **84.7%**. As the number of demonstration increases, RIPT-VLA consistently maintains a large advantage across all data regions. At 10 demonstrations, the average success rate of RIPT-VLA reaches **79.7%**, compared to only **29.4%** for SFT.

These results further show the limitation of SFT paradigm for VLA generalization under low-data regime. In contrast, we show that RIPT-VLA is not only help adapt pretrained skills to new environments, but also excels in fast generalization of task goal semantics.

## A.3 ADITIONAL STUDY

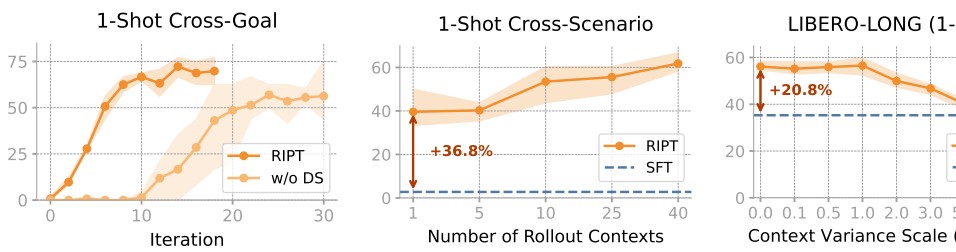

Figure 5: Training curve analysis of dynamic sampling.

Figure 6: Analysis on context dataset size.

Figure 7: Analysis on initial state std scale.

| Method | Goal | Spatial | Object | Long | 90 | ML45 | Average |
|---|---|---|---|---|---|---|---|
| QueST | 80.8 | 87.4 | 93.6 | 68.8 | 88.6 | 91.0 | 85.0 |
| + RIPT-VLA w/o Dynamic Sampling | 90.6 | 91.3 | 97.5 | 78.3 | 92.2 | 91.3 | 90.2 |
| **+ RIPT-VLA (Ours)** | **92.7** | **95.6** | **98.4** | **87.5** | **94.3** | **92.2** | **93.5** |

Table 3: Ablation on dynamic sampling. We compare full RIPT-VLA against a variant without dynamic sampling and the QueST baseline across task types and multitask suites.

**Effect of dynamic rollout sampling.** We ablate the impact of our dynamic rollout sampling strategy described in Section 4.2. We compare the full RIPT-VLA method with a variant that disables dynamic rejection. As shown in Table 3, dynamic sampling significantly boosts performance across all task categories and suites. By filtering out uninformative rollout groups, dynamic sampling ensures stable and efficient learning with consistent gradient signal across batches. On average, we observe a **+3.3** absolute improvement in success rate compared to the non-dynamic variant, demonstrating its crucial role in stabilizing RIPT-VLA training. In Figure 5, we show training curve (averaged over 3 seeds) of Column 2 of Figure 4. We see that dynamic rollout sampling accelerates convergence of RIPT-VLA, achieving consistently higher performance and more stable optimization.

**Effect of context dataset size.** To study how the size of the context dataset $\mathcal{D}_c$ impacts performance, we fix the QueST model SFT-trained with only 1 demonstration for Column 2 of Figure 3 and vary the number of rollout contexts used in the RIPT-VLA stage. As shown in Figure 6, increasing the number of rollout contexts significantly improves performance. This is because more contexts provide greater diversity in initial states for the rollouts interaction, allowing the model to better generalize across different setups in the testing environments. Notably, expanding $\mathcal{D}_c$ requires no additional human annotations: each context only consists of the initial observation state and no action is needed. This makes context dataset scaling a cost-effective way to enhance generalization of RIPT-VLA.

**Effect of context variance in RLOO group.** In Equation 2, each batch of rollouts is grouped by shared initial state contexts. In realistic deployments, however, perfectly matching initial states is impractical due to inevitable setup noise. To simulate this, we compute the standard deviation of object initial positions across LIBERO-LONG, which is around 2.5 cm. Starting with a QueST model SFT on 1 demo, we run `RIPT-VLA` while injecting Gaussian noise into the initial states with increasing scales of std. As shown in Figure 7, performance remains stable up to 1.0× (2.5 cm), and only begins to degrade beyond 2.0×. Remarkably, even at 7.0× variance (17.5 cm), `RIPT-VLA` still outperforms the SFT baseline by a significant margin.

## A.4 LLM USAGE

We used large language models (LLMs) primarily to improve the presentation and clarity of this paper. Specifically:

- Formatting of result tables in latex from CSV files. We ensure all numbers rigorously cross-checked against experiment outputs.
- Adjustments to the visual style of curve plots. We ensure all values match the underlying experimental results.
- Refinement of grammar and wording in the Abstract and Introduction to improve readability.

All experimental design, implementation, and analysis were conducted by the authors.

