# OpenReview forum: "Interactive Post-Training for Vision-Language-Action Models"
_ICLR.cc/2026/Conference — Submitted to ICLR 2026_

### Official Review · Reviewer_PkAa · 2025-10-27

**Soundness:** 2
**Presentation:** 2
**Contribution:** 2
**Rating:** 2
**Confidence:** 4

**Summary:**

This paper proposes RIPT-VLA, advancing VLA models via enabling interactive post-training with dynamic rollout sampling and stable reinforcement learning methods.
Experiments on various VLA benchmarks demonstrate that RIPT-VLA outperforms existing methods and has significant generalisation ability.

**Strengths:**

This paper presents a simple yet efficient training recipe for VLA models, including RLOO[1] and Dynamic sampling[2].


>reference
>>
>> [1] Buy 4 reinforce samples, get a baseline for free!
>>
>> [2] DAPO: An Open-Source LLM Reinforcement Learning System at Scale.

**Weaknesses:**

1. The main concern is the lack of novelty. Dynamic sampling is commonly used in LLMs, such as DAPO [1], which is not cited properly.
2. The citation of RLOO is wrong! It should be [1], but using [2] in this paper.
3. For VLA models, the experiments in real robotics should be considered.


>reference
>>
>> [1] DAPO: An Open-Source LLM Reinforcement Learning System at Scale.
>>
>> [2] Buy 4 reinforce samples, get a baseline for free!
>>
>> [3] Attention, Learn to Solve Routing Problems!

**Questions:**

See weakness.

---

> ### Author Response · Authors · 2025-11-24
> **Thank you for your helpful feedbacks!**
>
> We thank the reviewer for their helpful feedback. We address each of the points below:
>
> ## 1. Novelty of Dynamic Sampling
>
> Thank you for pointing this out. After reading DAPO again carefully we agree that the two sampling schemes are equivalent for 0/1 rewards. Our formulation might be more general for arbitrary rewards (i.e., advantage = 0 implies all rewards are the same and not just all 0 or 1as in DAPO) but we will clearly discuss and cite DAPO for the idea nonetheless.
>
> To the best of our knowledge, we are the first to train VLAs using this kind of binary verifier-based RL algorithm. We move beyond the standard "Pretraining + SFT" pipeline to introduce a scalable "Stage-3: Reinforcement Interactive Post-Training." Our extensive experiments prove that large-scale VLAs can be effectively fine-tuned with sparse binary rewards, a capability that was previously underexplored and is critical for scaling robotic learning without dense human supervision. We hope this actually leads to more VLAs being trained in interactive environments.
>
> ---
>
> ## 2. Incorrect RLOO Citation
>
> We appreciate the correction regarding the RLOO citation. We are sorry that this is a typo we had during drafting. We will update the paper to cite the correct paper pointed out.
>
> ---
>
> ## 3. Real-Robot Deployment for VLA Models
>
> Thank you for your comment. We would like to clarify that our primary deployment plan is to conduct the Stage-3 RIPT training completely in simulation before transferring the policy to the real world. This workflow directly alleviates the challenges raised by the reviewer as in simulation environment reset is precise and rollouts are computationally cheap.
>
> We decided to focus on LIBERO and ML45 was driven by the need for rigorous and reproducible comparisons using multiple state-of-the-art baselines (OpenVLA, QueST) in a controlled setting. While we currently lack access to the specific real robots  for these suites, we emphasize that improvements on these benchmarks are highly predictive of real-world performance. Recent VLA research has demonstrated that policies refined in simulation can successfully transfer to physical robots in a zero-shot manner [1,2,3,4,5]. Therefore, RIPT serves as a crucial pre-deployment optimization stage, robustifying policies in simulation to minimize the need for expensive real-world fine-tuning.
>
> Since our submission, recent follow-up work $\pi^{*}_{0.6}$ [6] has explicitly validated our interactive post-training paradigm on physical robots. They successfully deployed a similar "Stage-3" RL refinement workflow to solve complex real-world tasks (e.g., laundry folding). Crucially, they address the signal reliability issue by incorporating a strong value function learning process, demonstrating that our workflow adapts to the real world.
>
> ---
>
> ## **References**
>
> [1] Abouzeid et al. GeoAware-VLA: Implicit Geometry Aware Vision-Language-Action Model. arXiv:2509.14117.
>
> [2] Qu et al. SpatialVLA: Exploring Spatial Representations for Visual-Language-Action Model. arXiv:2501.15830.
>
> [3] Chen et al. InternVLA-M1: A Spatially Guided Vision-Language-Action Framework for Generalist Robot Policy. arXiv:2510.13778.
>
> [4] Wang et al. UniVLA: Unified Vision-Language-Action Model. arXiv:2506.19850.
>
> [5] Fang et al. ReBot: Scaling Robot Learning with Real-to-Sim-to-Real Robotic Video Synthesis. arXiv:2503.14526.
>
> [6] Physical Intelligence. π0.6∗​: a VLA That Learns From Experience. arXiv:2511.14759.

---

> > ### Comment · Reviewer_PkAa · 2025-11-27
> >
> > Thank you for your response. However, I maintain my original score. I find the novelty to be limited, particularly concerning the dynamic sampling technique. Furthermore, I believe that deploying the VLA model on a real robot is essential for a comprehensive evaluation.

---

### Official Review · Reviewer_ZxRz · 2025-11-01

**Soundness:** 2
**Presentation:** 3
**Contribution:** 2
**Rating:** 4
**Confidence:** 4

**Summary:**

This paper proposes RIPT-VLA, a post-SFT training stage for VLA models. It employs a critic-free RL
algorithm to fine-tune policies using sparse, binary environmental rewards, aiming to improve
performance in low-data regimes. The method is shown to improve performance significantly
over SFT-only baselines on several simulated benchmarks.

**Strengths:**

The main strength of this paper is showing that a simple, critic-free RL algorithm can effectively
fine-tune VLA models from sparse binary rewards. This is a practical and stable approach, and the
results convincingly show its ability to boost performance, especially when expert demonstrations
are scarce.

**Weaknesses:**

1. The main contribution of this paper is the dynamic sampling strategy, which, while effective,
is more of an engineering improvement than a fundamentally new RL algorithm. The novelty
lies more in the successful application and adaptation of this critic-free paradigm to the VLA
domain, rather than in inventing the RL method itself.

2. All experiments are conducted in simulation. While this is standard practice, policies trained
with RL can be sensitive to the sim-to-real gap. The paper claims the method is "practical,"
but does not discuss the potential challenges of transferring an interactively trained policy to
a physical robot, where interaction is costly and system dynamics may differ.

3. The paper highlights data efficiency in terms of the number of expert demonstrations.
However, it doesn't quantify the interaction cost required by the RL stage (i.e., the total
number of environment steps or wall-clock time needed for rollouts). This is a critical metric
for any RL method, as online interaction is often the real bottleneck, far more so than offline
SFT data.

**Questions:**

1. The authors should quantify the interaction cost. How many total environment steps (or wallclock
hours) were required for RIPT-VLA to converge? How does this "online" cost compare to
the "offline" cost of collecting the SFT data for the baselines?

2. The dynamic sampling strategy filters out contexts that are either "too easy" (all successes)
or "too hard" (all failures). Is there a risk of catastrophic forgetting on the "easy" tasks that
the policy no longer sees? Have the authors evaluated performance on these filtered-out
contexts post-training?

3. How sensitive is the method to K (number of rollouts per context)? The paper uses K=8/16.
What happens if K is very small (e.g., 2), which might be more practical on a real robot?

4. What new challenges do you foresee in applying RIPT-VLA to a real robotic system? For
instance, how would the algorithm handle noisy state estimation, physical safety constraints,
or the impossibility of perfectly resetting the environment?

---

> ### Author Response · Authors · 2025-11-24
> **Thank you for your helpful feedbacks! Response [1/2]**
>
> We thank the reviewer for their detailed and helpful feedback! We address each of the points below:
>
> ---
>
> ## 1. Algorithm Novelty
>
> > The main contribution of this paper is the dynamic sampling strategy, which, while effective, is more of an engineering improvement than a fundamentally new RL algorithm. The novelty lies more in the successful application and adaptation of this critic-free paradigm to the VLA domain, rather than in inventing the RL method itself.
>
> Thanks for your comments! To the best of our knowledge, we are the first to train VLAs using this kind of binary verifier-based RL algorithm. We move beyond the standard "Pretraining + SFT" pipeline to introduce a scalable "Stage-3: Reinforcement Interactive Post-Training." Our extensive experiments prove that large-scale VLAs can be effectively fine-tuned with sparse binary rewards, a capability that was previously underexplored and is critical for scaling robotic learning without dense human supervision. We hope this actually leads to more VLAs being trained in interactive environments. Since this submission to ICLR, follow-up work actually made this idea work in real robots [1].
>
> ---
>
> ## 2. Real-world Application
>
> > All experiments are conducted in simulation. While this is standard practice, policies trained with RL can be sensitive to the sim-to-real gap. The paper claims the method is "practical," but does not discuss the potential challenges of transferring an interactively trained policy to a physical robot, where interaction is costly and system dynamics may differ.
>
> Thank you for your comment. We would like to clarify that our primary deployment plan is to conduct the Stage-3 RIPT training completely in simulation before transferring the policy to the real world. This workflow directly alleviates the challenges raised by the reviewer as in simulation environment reset is precise and rollouts are computationally cheap.
>
> We decided to focus on LIBERO and ML45 was driven by the need for rigorous and reproducible comparisons using multiple state-of-the-art baselines (OpenVLA, QueST) in a controlled setting. While we currently lack access to the specific real robots for these suites, we emphasize that improvements on these benchmarks are highly predictive of real-world performance. Recent VLA research has demonstrated that policies refined in simulation can successfully transfer to physical robots in a zero-shot manner [2,3,4,5,6]. Therefore, RIPT serves as a crucial pre-deployment optimization stage, robustifying policies in simulation to minimize the need for expensive real-world fine-tuning.
>
> ---
>
> ## 3. Cost of Online Interaction
>
> > The paper highlights data efficiency in terms of the number of expert demonstrations. However, it doesn't quantify the interaction cost required by the RL stage… How many total environment steps or wall-clock hours were required? How does this compare to offline SFT?
>
> SFT is bottlenecked by active human supervision (teleoperation). RIPT shifts this burden to autonomous computers. The only human involvement is environment resetting, which is significantly cheaper and requires less expertise than collecting expert demonstrations. In Appendix A.3, we demonstrate that RIPT is highly robust to "imperfect resets" (up to 17.5cm initial state noise). This directly addresses the practicality of real-world resets, showing that high-precision laboratory setups are not required.
>
> Furthermore, the interaction cost is only a bottleneck if training occurs on the real robot. RIPT is ideally suited for improving policies in simulation (where interaction is free) prior to Sim-to-Real transfer. By using RIPT to maximize performance in simulation first, we reduce the reliance on expensive real-world data collection.
>
> To quantify, RIPT-VLA typically converges within 30 iterations for low-data tasks. With a batch size of ( B = 192 ) contexts and ( K = 8 ) rollouts per context, this totals ~46,080 episodes. While this would be substantial for a single physical robot, it is highly efficient for interaction in simulation, where rollouts can be parallelized across thousands of environments. Compared to standard RL methods that often require millions of samples to learn from scratch, RIPT is sample-efficient.
>
> ---
>
> ## Continue in the next post [2/2]

---

> ### Author Response · Authors · 2025-11-24
> **Thank you for your helpful feedbacks! Response [2/2]**
>
> ## 4. Dynamic Sampling & Catastrophic Forgetting
>
> > The dynamic sampling strategy filters out contexts that are either "too easy" or "too hard." Is there a risk of catastrophic forgetting? Did the authors evaluate performance on filtered-out contexts?
>
> Thank you for your discussion: this is an interesting point! The intuition you have here is right: it is true that all RL algorithms have a gradient of 0 if the advantage is zero. If the task is extremely hard/easy the algorithm does not consider that scenario. This is a property of all RL algorithms. Our filtering does not change this dynamic; it simply constructs batches out of samples that have non-zero gradients.
>
> Specifically, dynamic sampling does not discard easy/hard tasks. Hard contexts that initially result in 100% failure are filtered temporally. As the policy improves in easier contexts and generalization improves, the model eventually achieves some success on the harder tasks. This introduces variance (some fail, some succeed), causing the task to naturally re-enter the training buffer. This acts as an implicit curriculum rather than an optimization bias.
>
> If the policy begins to degrade on a "filtered" (easy) task, the success rate will drop below 100%. This immediately reintroduces variance into the reward signal. Consequently, the task automatically re-enters the training batch, allowing the policy to correct itself.
>
> ---
>
> ## 5. Effect of (K)
>
> > How sensitive is the method to K? What happens if K is small (e.g., 2)?
>
> To quantify the sensitivity, we conducted an ablation study on the LIBERO-Goal suite with varying group sizes ( K ). The results are summarized below:
>
> | Setting          | Success Rate (%) | Improvement over SFT |
> | ---------------- | ---------------- | -------------------- |
> | **SFT Baseline** | 80.8%            | —                    |
> | **RIPT (K=2)**   | 83.8%            | +3.0%                |
> | **RIPT (K=4)**   | 86.0%            | +5.2%                |
> | **RIPT (K=8)**   | **92.7%**        | **+11.9%**           |
>
> We note that RIPT still yields improvements at small ( K ) (( K = 2/4 )), although the performance gain is larger at ( K = 8 ). Note that since our workflow leverages simulation to build a strong policy before transfer (as noted in Weakness 2), the computational cost of using a larger ( K ) is not critical.
>
> ---
>
> ## 6. Challenges for Real-world Deployment
>
> > What new challenges arise when applying RIPT-VLA to a real robot? Noisy sensors, imperfect resets, safety constraints, etc.?
>
> We anticipate three main challenges for real-world deployment, which we will explicitly discuss in the final paper:
>
> **Environment resets:** Real-world resets are noisy and difficult to automate. While our Appendix A.3 demonstrates that RIPT is robust to initial state noise up to 17.5cm, we acknowledge that this tolerance might still be insufficient for highly unstructured or chaotic real-world environments where state variation exceeds this margin.
>
> **Reward estimation:** We utilize binary rewards specifically because they are robustly detectable via VLMs or simple sensors, avoiding the dense reward engineering that makes other RL methods impractical. However, this process can lead to noisy estimation which needs to be taken account.
>
> **Sim-to-real transfer:** When using RIPT in simulation to prepare for transfer, there is a risk that the policy may overfit to simulation artifacts (e.g., inaccurate physics approximations) that do not transfer well to the real world.
>
> We will add a dedicated "Practical Considerations" section to the final paper that explicitly outlines these potential pitfalls and discusses mitigation strategies.
>
> ---
>
> ## References
>
> [1] Physical Intelligence. π0.6∗​: a VLA That Learns From Experience. arXiv:2511.14759.
>
> [2] Abouzeid et al. GeoAware-VLA: Implicit Geometry Aware Vision-Language-Action Model. arXiv:2509.14117.
>
> [3] Qu et al. SpatialVLA: Exploring Spatial Representations for Visual-Language-Action Model. arXiv:2501.15830.
>
> [4] Chen et al. InternVLA-M1: A Spatially Guided Vision-Language-Action Framework for Generalist Robot Policy. arXiv:2510.13778.
>
> [5] Wang et al. UniVLA: Unified Vision-Language-Action Model. arXiv:2506.19850.
>
> [6] Fang et al. ReBot: Scaling Robot Learning with Real-to-Sim-to-Real Robotic Video Synthesis. arXiv:2503.14526.

---

> > ### Comment · Reviewer_ZxRz · 2025-11-28
> >
> > Thank you for the responses, which address my concerns regarding the soundness and practicality of the method. However, I still find the algorithmic novelty to be limited

---

### Official Review · Reviewer_1KyZ · 2025-11-03

**Soundness:** 3
**Presentation:** 3
**Contribution:** 3
**Rating:** 4
**Confidence:** 4

**Summary:**

The paper proposes RIPT-VLA, an interactive post‑training stage that follows pretraining and supervised fine‑tuning for vision‑language‑action models. In Stage 3 the policy is rolled out in a multitask environment and receives only binary success or failure rewards. The update rule is a critic‑free variant of PPO that couples leave‑one‑out advantages with importance‑weighted updates and a simple dynamic sampling rule that discards rollout groups whose advantages are all zero, which the authors argue stabilizes training when tasks are either trivially solved or always fail. The method applies to both tokenized action heads and continuous regression heads by adding a lightweight scale head so log probabilities are available for PPO. Algorithm 1 and Section 4 detail the training loop and the dynamic sampling design.

Experiments are entirely in simulation on LIBERO suites and Meta‑World 45. On Stage‑2 small models, RIPT gives larger absolute gains, for example QueST improves from 82.7 to 93.6 average success across LIBERO Goal, Spatial, Object, Long and sees 18 points on LIBERO‑Long. On a strong Stage‑1+2 large model, OpenVLA‑OFT, gains are modest in absolute terms, from 96.7 to 97.5 average. The paper also reports many‑task results on LIBERO‑90 and ML45, few‑shot SFT regimes, and cross‑scenario or cross‑goal generalization curves in the appendix. Table 1 on page 8 and Table 2 on page 9 summarize the core numbers, Figure 2 on page 9 shows the few‑shot curve, and Figures 3-4 in the appendix show generalization plots. The ethics statement explicitly notes that all experiments are in simulation only.

**Strengths:**

• Clear, simple recipe for interactive post‑training after SFT. Algorithm and the dynamic sampling heuristic are easy to implement. Section 4 and Algorithm 1 are well written.

• Some gains in the areas that matter most for data scarcity. On LIBERO‑Long and few‑shot settings, the improvements over SFT are moderate for Stage 2 models, and the “1‑demo to workable policy” result is an interesting result if generalizable.  Certainly gains are shown, though only one (seemingly easy) example was shown in the paper.

• The method covers both tokenized and continuous action heads by adding a small scale head for regression policies, which increases applicability across current VLA families. Section 4.3 describes this clearly.

• Using RL as a third stage generally seems like a good idea idea, though a binary signal might be challenging to apply broadly.

**Weaknesses:**

• Binary reward is probably too constraining for many real settings and it's perhaps interesting to used a learned reward signal.

• External validity. All results are in simulators, with LIBERO and ML45, and the paper does not demonstrate the method on a physical robot or discuss how robust success detectors would be implemented in hardware. For example, how might you replay an exact context?  This would need to include the environment state, so you'd have to do environmental resets, which seems impractical. Real‑robot experiments or a concrete plan for doing this in the real world would be important for impact for a paper like this.

• Limited absolute gains on a strong Stage‑1+2 baseline. In Table 1 the OpenVLA‑OFT average moves from 96.7 to 97.5. This suggests diminishing returns when the model already encodes broad world knowledge, which the paper should acknowledge more directly in the discussion. An analysis of this would help readers decide when Stage‑3 is worth the extra rollouts.

• It would be nice to know how this might take us to a world of generalization beyond just a per-task SFT.  While I understand this is what the community benchmarks on, I'd hope a more general RL framework like this might also enable generality across tasks as well.

• Dynamic sampling discards groups with all‑zero or all‑one advantages. This can bias learning away from the hardest contexts that currently always fail and from already‑solved but still informative contexts.  It would be nice to still learn from these examples.

• Comparison to “more SFT” is missing. Since Stage‑3 is an extra training phase, it ought to beat a carefully matched alternative that collects a small number of additional SFT demonstrations. I don't believe I saw this in the paper?  The few‑shot curves show that more SFT helps, but there is no direct human‑effort trade‑off between X additional demos and Y Stage‑3 rollout steps. It would be nice to add a controlled study that matches person‑hours  or data collection time or just raw samples.

Small things:
Abstract: leave-on-out -> "leave-one-out"

**Questions:**

* How would you construct reliable success signals in the real world for the LIBERO‑like tasks and how sensitive is RIPT to false labels? Do you expect dynamic sampling to amplify errors when detectors are noisy?  While the paper presents interesting results in simulation, I'm concerned that this approach is pretty impractical to apply in the real world (major concern).  Can you comment on this?

* What is the sample complexity of Stage‑3 relative to adding demonstrations? It would be nice to compare against just adding a few more SFT examples.

* For the continuous head, does training the scale head for NLL change the original OFT policy’s action distribution or degrade performance before RL begins?

---

> ### Author Response · Authors · 2025-11-24
> **Thank you for your helpful feedbacks! Response [1/3]**
>
> We thank the reviewer for their detailed feedback and for acknowledging our method's effectiveness in simulation. We appreciate the opportunity to clarify our deployment process and the practical implications of our work. We response point-by-point below:
>
> ## 1. External Validity and Read-World Practice
>
> > Binary reward is probably too constraining for many real settings and it's perhaps interesting to used a learned reward signal.
>
> > External validity. All results are in simulators, with LIBERO and ML45, and the paper does not demonstrate the method on a physical robot or discuss how robust success detectors would be implemented in hardware. For example, how might you replay an exact context? This would need to include the environment state, so you'd have to do environmental resets, which seems impractical. Real-robot experiments or a concrete plan for doing this in the real world would be important for impact for a paper like this.
>
> *Binary Rewards*: We argue that binary rewards are more practical for scaling than learned reward signals. Implementing a binary success detector (e.g., "is the drawer open?") via even general-purpose Vision-Language Models is significantly easier than engineering the dense reward functions or training reward models that require significant human labels and suffer from issues like reward hacking.
>
>
> *Real-world Experiments*: Thank you for your comment. We would like to clarify that our primary deployment plan is to conduct the Stage-3 RIPT training completely in simulation before transferring the policy to the real world. This workflow directly alleviates the challenges raised by the reviewer as in simulation environment reset is precise and rollouts are computationally cheap.
>
> We decided to focus on LIBERO and ML45 was driven by the need for rigorous and reproducible comparisons using multiple state-of-the-art baselines (OpenVLA, QueST) in a controlled setting. While we currently lack access to the specific real robots  for these suites, we emphasize that improvements on these benchmarks are highly predictive of real-world performance. Recent VLA research has demonstrated that policies refined in simulation can successfully transfer to physical robots in a zero-shot manner [1,2,3,4,5]. Therefore, RIPT serves as a crucial pre-deployment optimization stage, robustifying policies in simulation to minimize the need for expensive real-world fine-tuning. Since our submission, recent follow-up work $\pi^{*}_{0.6}$ [6] has explicitly validated our interactive post-training paradigm on physical robots.
>
> ---
>
> ## 2. Gains on OpenVLA:
> > Limited absolute gains on a strong Stage-1+2 baseline. In Table 1 the OpenVLA-OFT average moves from 96.7 to 97.5. This suggests diminishing returns when the model already encodes broad world knowledge, which the paper should acknowledge more directly in the discussion. An analysis of this would help readers decide when Stage-3 is worth the extra rollouts.
>
> While the absolute gain on OpenVLA-OFT appears small (96.7% -> 97.5%), this represents a 24% reduction in failure rate (3.3% -> 2.5%), which is critical for high-reliability systems. More importantly, one of RIPT-VLA's main contributions lies in the low-data regime. As highlighted in Figure 1 and Table 2, RIPT transforms an unusable 1-shot SFT model (4% success) into a state-of-the-art model (97% success). We believe this capability of making models useful in data-scarce environments is where Stage-3 training adds the most value.
>
> ---
>
> ## 3. Generalization across task
>
> > It would be nice to know how this might take us to a world of generalization beyond just a per-task SFT. While I understand this is what the community benchmarks on, I'd hope a more general RL framework like this might also enable generality across tasks as well.
>
> Firstly, we would like to clarify that our main evaluations are not per-task SFT. As detailed in Section 5.2 and Table 2, our results on LIBERO-90 and MetaWorld-45 (ML45) involve training a single VLA model to solve 90 and 45 distinct tasks simultaneously. RIPT-VLA successfully improves this global multi-task policy (e.g., +5.7% on LIBERO-90), demonstrating its effectiveness in large-scale multi-task settings.
>
> We agree that RL should enable broader cross-task generalization to unseen tasks during large-scale training. We investigated this in Appendix A.1 (Cross-Scenario) and Appendix A.2 (Cross-Goal). We found that RIPT-VLA enables a model trained on one scenario to generalize to a new, unseen scenario with only 1-5 demonstrations, boosting success from ~5% (SFT) to >90% (RIPT). This demonstrates that RIPT helps the model adapt its pre-trained priors to new contexts far more effectively than SFT.
>
> ---
>
> ## Continue in the next post [2/3].

---

> ### Author Response · Authors · 2025-11-24
> **Thank you for your helpful feedbacks! Response [2/3]**
>
> ---
>
> ## 4. Dynamic Sampling & Catastrophic Forgetting
>
> > Dynamic sampling discards groups with all-zero or all-one advantages. This can bias learning away from the hardest contexts that currently always fail and from already-solved but still informative contexts. It would be nice to still learn from these examples.
>
> Thank you for your discussion: this is an interesting point! The intuition you have here is right: it is true that all RL algorithms have a gradient of 0 if the advantage is zero. If the task is extremely hard/easy the algorithm does not consider that scenario. This is a property of all RL algorithms. Our filtering does not change this dynamic; it simply constructs batches out of samples that have non-zero gradients.
>
> Specifically, dynamic sampling does not discard easy/hard tasks. Hard contexts that initially result in 100% failure are filtered temporally. As the policy improves in easier contexts and generalization improves, the model eventually achieves some success on the harder tasks. This introduces variance (some fail, some succeed), causing the task to naturally re-enter the training buffer. This acts as an implicit curriculum rather than an optimization bias. If the policy begins to degrade on a "filtered" (easy) task, the success rate will drop below 100%. This immediately reintroduces variance into the reward signal. Consequently, the task automatically re-enters the training batch, allowing the policy to correct itself.
>
> ---
>
> ## 5. Comparison to “More SFT”.
>
> > Comparison to “more SFT” is missing. Since Stage-3 is an extra training phase, it ought to beat a carefully matched alternative that collects a small number of additional SFT demonstrations. I don't believe I saw this in the paper? The few-shot curves show that more SFT helps, but there is no direct human-effort trade-off between X additional demos and Y Stage-3 rollout steps. It would be nice to add a controlled study that matches person-hours or data collection time or just raw samples.
>
> We address the trade-off between "more SFT" and "Stage-3 RL" in Figure 2. The results show that 1-shot RIPT (~ 97% success) significantly outperforms even 10-shot SFT (~ 60% success).
>
> Trader-offs are hard to calibrate. If we use SFT, it requires a human demonstrator to specify a trajectory via teleoperation, which demands skilled labor, continuous attention, and specialized equipment. This is very different from the supervision that RL would need (essentially we just need to write a verifier). We will mention in our paper that in our case writing this binary verifier is easy, because the tasks we consider have clear success conditions and can be written with simulation rules easily. Maybe for more complex tasks (like multi-step tasks or with subjective goals) this verifier might be harder to obtain.
>
> Another issue is reset: both SFT and RIPT rely on resets. In SFT this can be combined with demonstration; in RL this needs to be done by human effort if not in simulation (our paper mainly focuses on training in simulation). But overall the direct comparison between the two along a single metric is tricky.
>
> ---
>
> ## Continue in the next post [3/3].

---

> ### Author Response · Authors · 2025-11-24
> **Thank you for your helpful feedbacks! Response [3/3]**
>
> ## 6. Sensitivity to binary label noise.
>
> > How would you construct reliable success signals in the real world for the LIBERO‑like tasks and how sensitive is RIPT to false labels? Do you expect dynamic sampling to amplify errors when detectors are noisy? While the paper presents interesting results in simulation, I'm concerned that this approach is pretty impractical to apply in the real world (major concern). Can you comment on this?
>
> Thanks for the question, this is a valid concern. Since our submission, recent follow-up work $\pi^{*}_{0.6}$ [6] has explicitly validated our interactive post-training paradigm on physical robots. They successfully deployed a similar "Stage-3" RL refinement workflow to solve complex real-world tasks (e.g., laundry folding). Crucially, they address the signal reliability issue by incorporating a strong value function learning process, demonstrating that our workflow adapts to the real world. We hope our  work inspires more VLA models to be trained in interactive environments.
>
> We do not expect Dynamic Sampling to amplify errors significantly. First, RL algorithms like PPO are generally robust to zero-mean noise in the reward signal due to batch aggregation ($B=192$ in our setup). Second, if a detector is noisy and flags a failure on a "solved" task, Dynamic Sampling will retain that batch (detecting variance>0). Dynamic sampling only ignores the all-success/all-failure contexts (which is very rare when there is label noise); and even when that happens, Dynamic sampling only ignores them in the current optimization step and will continue to consider them in the next round.
>
> ---
>
> ## 7. Sample Complexity;
>
> > What is the sample complexity of Stage‑3 relative to adding demonstrations? It would be nice to compare against just adding a few more SFT examples.
>
> We argue that comparing raw "sample counts" might not be as informative because the type of effort differs fundamentally. SFT requires active human teleoperation, which is expensive, tiring, and hard to scale. In contrast, RIPT moves the burden to autonomous machine computation. The human cost in RIPT is environment resetting, which is significantly cheaper and does not require expert skills.
>
> Furthermore, the high sample complexity of RL can be addressed by running RIPT entirely in simulation to perfect the policy before deployment. Since RIPT significantly boosts robustness and success rates in simulation, the resulting policy provides a much stronger initialization for the real world compared to SFT, easing the subsequent Sim-to-Real domain transfer. While a full Sim-to-Real study is beyond the scope of this work, RIPT enables this scalable workflow, whereas SFT is bound by the bottleneck of manual human data collection.
> We will add a dedicated "Practical Considerations" section to the final paper that explicitly outlines these potential pitfalls and discusses mitigation strategies, ensuring readers can apply the method with a clear understanding of these transfer risks.
>
> ---
>
> ## 8. Performance effect of the scale head:
>
> > For the continuous head, does training the scale head for NLL change the original OFT policy’s action distribution or degrade performance before RL begins?
>
>
> Since we fix the original model (backbone and mean action head), the policy's action predictions ($\mu$) remain mathematically identical to the original OFT policy. Therefore, training the scale head for NLL does not change the original OFT policy’s action mean distribution and the performance does not degrade if we use the action output by the mean action head.
>
> ---
>
> # References
> [1] Abouzeid et al. GeoAware-VLA: Implicit Geometry Aware Vision-Language-Action Model. arXiv:2509.14117.
>
> [2] Qu et al. SpatialVLA: Exploring Spatial Representations for Visual-Language-Action Model. arXiv:2501.15830.
>
> [3] Chen et al. InternVLA-M1: A Spatially Guided Vision-Language-Action Framework for Generalist Robot Policy. arXiv:2510.13778.
>
> [4] Wang et al. UniVLA: Unified Vision-Language-Action Model. arXiv:2506.19850.
>
> [5] Fang et al. ReBot: Scaling Robot Learning with Real-to-Sim-to-Real Robotic Video Synthesis. arXiv:2503.14526.
>
> [6] Physical Intelligence. π0.6∗​: a VLA That Learns From Experience. arXiv:2511.14759.

---

### Meta-Review · Area_Chair_5WFJ · 2025-12-06

**Summary:**

**Paper Summary**

This paper proposes RIPT-VLA, a VLA framework that introduces a new interactive post-training phase with an RL objective to enhance model performance during inference. A dynamic-sampling strategy is also presented to make this third training stage more stable and efficient. Experiments on LIBERO and Meta-World demonstrate that RIPT-VLA outperforms state-of-the-art VLA models trained with SFT. The paper concludes that the interactive post-training phase is effective for post-training VLA models.

---

After reading the paper and reviewers' comments, the AC summarizes the paper's strengths and weaknesses below.

**Strengths**
- The framework is easy to implement and straightforward.
- The framework demonstrates performance gains compared to VLA methods trained with SFT.
- The paper clearly presents its arguments and technical details.

**Weaknesses**
- The technical novelty of the proposed framework is limited.
- The proposed framework has not been deployed in real-world environments.
- Some design choices are not well-verified or introduced with convincing justification.

**Reviewer Concerns:**

After reviewing the author rebuttal, some weaknesses such as design choices are addressed by providing additional ablation studies. However, the AC believes the following issues remain unresolved:

- **Lacking real-world deployments or analysis**. This issue is raised by multiple reviewers. Although the authors argue that recent works have examined similar RL post-training strategies in real-world robot deployments, the AC believes the submission itself should at least conduct real-world demos or provide an in-depth discussion on which scenarios the RL post-training phase can be applied to. A similar strategy working in real-world environments does not guarantee that the proposed framework will also work. Interacting with real-world environments may cause severe damages, so it should be designed and examined carefully.

- **The technical novelty is limited**. This issue is also a consensus among reviewers. To be clear, it is fine to propose a simple yet effective solution, as long as it is first proposed and has been verified extensively. In the paper, the proposed framework is only tested with one VLA in each category (two in total), which is not sufficient. Moreover, the AC also feels that the comparison between adding the third RL post-training phase and adding more demonstrations for the SFT phase is too naive. The diversity and quality of demonstrations should also be considered, since the biggest advantage of RL post-training is that it can explore and retrieve knowledge that is not included in the original expert demonstration set.

**Reviewer Scores:**

The paper initially received scores of [2, 4, 4]. After the discussion phase, two reviewers indicated that at least one concern remains and decided to keep their scores. These concerns were also pointed out by the third reviewer in the preliminary review. As a result, the AC believes it is unlikely that the paper’s scores would increase even with a full author-reviewer discussion.

---

### Decision · Program_Chairs · 2026-01-26

Reject